# Uptake of COVID-19 Vaccines among Pregnant Women: A Systematic Review and Meta-Analysis

**DOI:** 10.3390/vaccines10050766

**Published:** 2022-05-12

**Authors:** Petros Galanis, Irene Vraka, Olga Siskou, Olympia Konstantakopoulou, Aglaia Katsiroumpa, Daphne Kaitelidou

**Affiliations:** 1Clinical Epidemiology Laboratory, Faculty of Nursing, National and Kapodistrian University of Athens, 15773 Athens, Greece; aglaiakat@nurs.uoa.gr; 2Department of Radiology, P. & A. Kyriakou Children’s Hospital, 15773 Athens, Greece; irenevraka@yahoo.gr; 3Center for Health Services Management and Evaluation, Faculty of Nursing, National and Kapodistrian University of Athens, 15773 Athens, Greece; olsiskou@nurs.uoa.gr (O.S.); olympiak1982@hotmail.com (O.K.); dkaitelid@nurs.uoa.gr (D.K.)

**Keywords:** pregnant women, COVID-19, vaccination, uptake, predictors

## Abstract

Mass vaccination against COVID-19 is essential to control the pandemic. COVID-19 vaccines are now recommended during pregnancy to prevent adverse outcomes. With this review, we aimed to evaluate the evidence in the literature regarding the uptake of COVID-19 vaccinations among pregnant women. A comprehensive search was performed in PubMed, Medline, Scopus, ProQuest, Web of Science, CINAHL, and medRxiv from inception to 23 March 2022. We performed a meta-analysis to estimate the overall proportion of pregnant women vaccinated against COVID-19. We found 11 studies including 703,004 pregnant women. The overall proportion of pregnant women vaccinated against COVID-19 was 27.5% (95% CI: 18.8–37.0%). Predictors of COVID-19 vaccination uptake were older age, ethnicity, race, trust in COVID-19 vaccines, and fear of COVID-19 during pregnancy. Mistrust in the government, diagnosis of COVID-19 during pregnancy, and fears about the safety and side effects of COVID-19 vaccines were reasons for declining vaccination. The global COVID-19 vaccination prevalence in pregnant women is low. A large gap exists in the literature on the factors influencing the decision of pregnant women to be vaccinated against COVID-19. Targeted information campaigns are essential to increase vaccine literacy among pregnant women.

## 1. Introduction

Pregnant women with COVID-19 are at increased risk of severe illness, adverse birth outcomes, and mortality. In particular, hospitalized pregnant women with symptomatic COVID-19 are more likely to have iatrogenic preterm births, be admitted to intensive care, and need invasive ventilation than pregnant women without COVID-19 [1,2,3,4,5]. For instance, in the United Kingdom, between February and September 2021, 98% of the 1714 pregnant women admitted to hospital with symptomatic COVID-19 were unvaccinated [6], while no fully vaccinated pregnant women were admitted to intensive care with COVID-19 [7]. Moreover, during pregnancy, pregnant women are prone to developing a higher susceptibility to viral infections, which might lead, in certain circumstances, to pregnancy complications, preterm births, and even miscarriages [8,9].

Pregnant women were not included in the initial randomized controlled trials testing COVID-19 vaccines, leading to a lack of data on vaccination safety and pregnancy outcomes compared with the general population [10,11]. However, two systematic reviews found that reactogenicity is similar in pregnant women and the general population, abortion rates are similar in vaccinated and nonvaccinated pregnant women studied before the COVID-19 pandemic, and anti-SARS-CoV-2 immunoglobulins are transferred through the placenta and the breast milk to newborns, providing protective immunity [12,13]. Moreover, according to a systematic review of studies in the USA, pregnant women have the same risk of adverse pregnancy or neonatal outcomes as unvaccinated pregnant women [14]. In general, COVID-19 vaccination produces immune responses during pregnancy and does not cause vaccine-related adverse events [15,16]. Thus, COVID-19 vaccines are commonly officially suggested for pregnant women. In particular, several organizations such as the Center for Disease Control, the Society for Maternal-Fetal Medicine, and the American College of Obstetricians and Gynecologists now recommend that pregnant women should receive COVID-19 vaccines to prevent severe maternal morbidity and adverse birth outcomes [17,18,19].

To the best of our knowledge, none of the previous systematic reviews provided evidence about the uptake of COVID-19 vaccines among pregnant women. Therefore, our aim in this systematic review was to identify what is known about the uptake of COVID-19 vaccines among pregnant women. We also investigated predictors of COVID-19 vaccination uptake among pregnant women and reasons for declining vaccination.

## 2. Materials and Methods

### 2.1. Data Sources and Strategy

We conducted a systematic review following the Preferred Reporting Items for Systematic Reviews and Meta-Analysis (PRISMA) guidelines [20]. We searched PubMed, Medline, Scopus, ProQuest, Web of Science, CINAHL, and a preprint service (medRxiv) from inception to 23 March 2022. We used the following strategy in all fields: ((pregnan*) AND (vaccin*)) AND (COVID-19). 

### 2.2. Selection and Eligibility Criteria

Three independent authors applied a three-step procedure for selecting studies: removal of duplicates, screening of title and abstract, and reading of full-text articles. Two independent authors performed study selection and a third senior author resolved any differences. Moreover, we examined the reference lists of all relevant articles. The population of interest was pregnant women, and the outcome was COVID-19 vaccination uptake. Thus, we included quantitative studies reporting COVID-19 vaccination uptake among pregnant women, studies that examined predictors of COVID-19 vaccination uptake, and studies that examined reasons for decline of vaccination. We included any study with information about COVID-19 vaccination uptake in pregnant women independent of the semester of pregnancy. Studies published in English were eligible for inclusion. We excluded reviews, protocols, posters, case reports, statements, letters to the editor, expert opinions, and editorials. 

### 2.3. Data Extraction and Risk of Bias Assessment

Three reviewers independently extracted the following data from the studies: authors, country, data collection time, sample size, age of pregnant women, study design, sampling method, response rate, percentage of COVID-19 vaccination uptake among pregnant women, predictors of COVID-19 vaccination uptake, reasons for declining COVID-19 vaccination, and type of publication (journal or preprint service). 

We used the Joanna Briggs Institute critical appraisal tool to assess risk of bias of the studies [21]. The response options are the following: Yes, when the criteria are clearly identifiable throughout the article; No, when the criteria are not identifiable; Unclear, when the criteria are not clearly identified in the article; and Not Applicable, when the criteria do not apply to the study. The risk of bias is ranked as “low”, “moderate”, or “high”, according to the percentage of “Yes” responses.

### 2.4. Statistical Analysis

The outcome variable was COVID-19 vaccination uptake among pregnant women. We divided the number of vaccinated pregnant women by the total number of pregnant women to calculate the proportion of pregnant women that was vaccinated against COVID-19. Then, we transformed this proportion with the Freeman–Tukey double arcsine method and we calculated the respective 95% confidence intervals (CI) for the proportions [22]. We used I2 and the Hedges Q statistics to assess heterogeneity between studies. An I2 value higher than 75% indicates high heterogeneity, and a *p*-value < 0.1 for the Hedges Q statistic indicates statistically significant heterogeneity [23]. Heterogeneity between results was very high; thus, we applied a random effect model to estimate the pooled proportion of COVID-19 vaccinated pregnant women [23]. We considered country, data collection time, sample size, age of pregnant women, study design, sampling method, response rate, risk of bias, and publication type (journal or pre- service) as prespecified sources of heterogeneity. Due to the scarce data and the high heterogeneity of the results of some variables (e.g., age of pregnant women), we decided to perform subgroup analysis for risk of bias, study design, and the country that studies were conducted in. We also performed meta-regression analysis using sample size and data collection time as the independent variables. We treated data collection time as a continuous variable, assigning the number 1 for studies that were conducted in December 2020, the number 2 for studies that were conducted in January 2020, etc. We performed a leave-one-out sensitivity analysis to estimate the influence of each study on the overall proportion of COVID-19 vaccinated pregnant women. We used a funnel plot and Egger’s test to assess publication bias. Regarding Egger’s test, a *p*-value < 0.05 indicates publication bias [24]. We did not perform a meta-analysis of the factors that affected the pregnant women’s decision to be vaccinated against COVID-19 because the data were very scarce. We used OpenMeta [Analyst] for the meta-analysis [25].

## 3. Results

### 3.1. Identification and Selection of Studies

A flowchart of our systematic review is shown in Figure 1. Our initial search yielded 6932 records after removing duplicates. Applying the inclusion and exclusion criteria, we identified 11 articles.

### 3.2. Characteristics of the Studies

We found 11 studies including 703,004 pregnant women. The main characteristics of the studies included in this review are presented in Table 1. Four studies were conducted in Israel [26,27,28,29], three studies in the USA [30,31,32], two studies in the United Kingdom [7,33], one study in Japan [34], and one study in Scotland [35]. Data collection time among studies ranged from December 2020 [30,31] to October 2021 [35]. Sample size ranged from 473 [32] to 355,299 pregnant women [7]. Eight studies were cohort studies [7,26,27,29,30,31,33,35] and three studies were cross-sectional [28,32,34]. Two studies used national data [7,35], three studies used a convenience sample [28,32,34], and six studies did not report the sampling method [26,27,29,30,31,33]. Ten studies were published in peer-reviewed journals [7,26,27,28,29,30,31,33,34,35] and one study was published in a preprint service [32].

### 3.3. Risk of Bias Assessment

The risk of bias assessment of studies included in this review is shown in Appendix A. The risk of bias was moderate in five cohort studies [26,27,29,30,35] and low in three cohort studies [7,31,33]. The most common bias in cohort studies was the absence of strategies to address incomplete follow up. Only one cohort study [33] used multivariate analysis to eliminate confounding. Regarding cross-sectional studies, the risk of bias was low in two studies [32,34] and moderate in one study [28]. 

### 3.4. COVID-19 Vaccination Uptake

The overall proportion of vaccinated pregnant women against COVID-19 was 27.5% (95% CI: 18.8–37.0%) (Figure 2). COVID-19 vaccination uptake among pregnant women ranged from 7.0% (95% CI: 6.9–7.1%) [7] to 68.7% (95% CI: 68.2–69.3%) [26]. The heterogeneity between results was very high (I2 = 99.98%, *p*-value for the Hedges Q statistic <0.001). The results of leave-one-out sensitivity analysis showed that no single study had a disproportional effect on the overall proportion, which varied between 23.5% (95% CI: 18.5–28.8%), with Goldshtein et al. (2022) [26] excluded, and 29.7% (95% CI: 18.9–41.9%) with UK Health Security Agency (2021) [7] excluded. Publication bias was probable according to Egger’s test (<0.05) and the funnel plot (Appendix A).

According to subgroup analysis, the pooled proportion of the studies with a moderate risk of bias (33.0% (95% CI: 13.8–55.8%), I2 = 99.99) was higher than that of the studies with a low risk of bias (21.0% (95% CI: 13.8–29.3%), I2 = 99.96). The type of study was another source of heterogeneity, because the pooled proportion of the cross-sectional studies (34.7% (95% CI: 11.9–62.1%), I2 = 99.52) was higher than that of the cohort studies (24.9% (95% CI: 15.3–35.9%), I2 = 99.99). Moreover, the pooled proportion of studies that were conducted in Israel (43.3% (95% CI: 17.1–71.8%), I2 = 99.93) was higher than those of studies that were conducted in the USA (27.3% (95% CI: 21.6–33.3%), I2 = 99.78) and other countries (12.8% (95% CI: 10.4–15.4%), I2 = 99.71). According to the results of meta-regression analysis, COVID-19 vaccination uptake among pregnant women was independent of sample size (*p* = 0.07) and data collection time (*p* = 0.34). 

### 3.5. Factors Related to COVID-19 Vaccination Uptake

Predictors of COVID-19 vaccination uptake among pregnant women and reasons for declining vaccination are shown in Table 2. Five studies investigated factors that affect pregnant women’s decision to vaccinate against COVID-19 [7,31,32,33,34]. Three studies [32,33,34] used multivariate analysis to eliminate confounding, and two studies [7,31] used descriptive statistics to present relationships between factors and COVID-19 vaccination uptake among pregnant women. 

Two studies [7,31] found that increased age was related to increased probability of COVID-19 vaccination uptake. Two studies [7,31] also found that White women and Asian women were vaccinated against COVID-19 more often than Black women and Hispanic women, whereas one study [7] found that vaccination was highest among women living in the least deprived areas and lowest among women living in the most deprived areas. Trust in COVID-19 vaccines, fear of COVID-19 during pregnancy, and pregestational diabetes mellitus were predictors of COVID-19 vaccination uptake among pregnant women [32,33]. Mistrust in the government, COVID-19 diagnosis during pregnancy, and fears about the safety and side effects of COVID-19 vaccines were reasons for declining vaccination [32,34].

## 4. Discussion

In this systematic review and meta-analysis, we estimated the COVID-19 vaccination uptake among pregnant women, and examined predictors of uptake of and reasons for declining vaccination. Eleven studies met our inclusion and exclusion criteria, and we found that worldwide the uptake prevalence of vaccination against COVID-19 was 27.5% in pregnant women. This prevalence is considerably lower than that of pregnant women who expressed the intention to be vaccinated against COVID-19. In particular, two meta-analyses [36,37] found that the global prevalence of pregnant women accepting the COVID-19 vaccine was about 49–54%. Moreover, in a survey with 5282 pregnant women from 16 countries, 52% of them indicated an intention to receive a COVID-19 vaccine [38]. 

Our review provides evidence of low levels of vaccine uptake in pregnant women. The proportion of pregnant women vaccinated against COVID-19 was even lower in studies with a low risk of bias and the cohort studies. Thus, our estimation is most likely an overestimation of the true global prevalence of pregnant women vaccinated against COVID-19 because the quality and type of study seem to have a significant impact on the results of the studies. Moreover, we found that the vaccination rate was much higher in Israel than in other countries. This large difference may be due to the fact that Israel was one of the first countries that launched a national vaccination project encouraging all pregnant women to receive a COVID-19 vaccine [39]. Notably, 4 of the 11 studies included in this review were conducted in Israel. This further demonstrates the urgency in Israel to inoculate the entire adult population, including pregnant women, as quickly as possible. 

The vaccine uptake rate did not improve even when data from studies began to demonstrate the safety and efficacy of COVID-19 vaccines in pregnant women [40,41]. However, the number of studies carried out since the publication of this information is very small and not sufficient to draw firm conclusions. 

Of the 11 studies in this review, 5 examined factors that are associated with COVID-19 vaccine uptake in pregnant women. Older pregnant women were positively associated with vaccine uptake. This finding has been confirmed by studies that investigated the intention of pregnant women to accept a COVID-19 vaccine. Several studies found that older age is related to higher acceptance of COVID-19 vaccines [38,42,43]. This finding is plausible because it is well-known that pregnancy at an advanced maternal age is a risk factor of adverse outcomes, such as higher rates of neonatal intensive care unit admission, preterm deliveries, spontaneous miscarriage, pre-eclampsia, low-birthweight babies, preterm labor, worse Apgar scores, and Cesarean deliveries [44,45]. Moreover, older age is associated with higher COVID-19 mortality [46,47,48]. It is probable that older pregnant women confront COVID-19 with more fear, resulting in their higher COVID-19 vaccination uptake [38,49]. 

According to our review, COVID-19 vaccination rate was highest among White and Asian pregnant women, and lowest among Black and Hispanic pregnant women. Hispanic ethnicity and Black or African American race are related to refusal of COVID-19 vaccination during pregnancy [31,42,50,51,52]. A systematic review found that White individuals have a higher rate of COVID-19 vaccine uptake than Black individuals [53]. Additionally, similar racial and ethnic disparities were reported for the acceptance of other recommended vaccinations during pregnancy, such as tetanus, influenza, and acellular pertussis, with Black and Hispanic women having the lowest vaccination coverage [54]. 

Trust in COVID-19 vaccines and fewer worries about the safety and side effects of COVID-19 vaccines are predictors of COVID-19 vaccination uptake. Similar factors, such as trust in the safety and efficacy of COVID-19 vaccines, confidence in received information on COVID-19 vaccination, not fearing COVID-19 vaccine side effects, trust in childhood vaccines, and influenza vaccination within the previous year, are associated with a higher intention rate of pregnant women to receive a COVID-19 vaccine [38,55,56,57,58]. In general, high levels of information and knowledge about COVID-19 vaccines decrease fear and have significant influence on a pregnant woman’s decision to receive COVID-19 vaccination. Moreover, a recent systematic review and meta-analysis found that COVID-19 vaccination protects pregnant women from SARS-CoV-2 infection and COVID-19-related hospitalization and does not have adverse events on pregnant women, fetuses, or newborns [59]. Because COVID-19 vaccines are proven to be safe and effective in pregnant women, policy makers should use this information to improve trust and confidence in COVID-19 vaccines and reduce hesitancy.

### Limitations

Our review and meta-analysis is subject to some limitations. Data taken from databases may not provide the most up-to-date evidence regarding COVID-19 vaccination uptake among pregnant women due to the publication process. This limitation is of particular importance in the present review, as the data on vaccination of pregnant women have been constantly increasing. Moreover, data collection time among studies ranged from December 2020 to October 2021, whereas evidence regarding safety and efficacy of COVID-19 vaccines in pregnant women has been significantly increasing on an ongoing basis. Thus, we should interpret the results of this review with care because they may not directly predict the future behavior of pregnant women. Additionally, we cannot generalize our results because the number of relevant studies included in this review is low and these studies were conducted only in five countries.

Only five studies examined the factors that affect pregnant women’s decisions to receive a COVID-19 vaccine. Moreover, these studies mainly investigated demographic factors, e.g., age, ethnicity, race, etc. The gap in the literature on the factors influencing the decision of pregnant women to be vaccinated against COVID-19 is large. For instance, psychological factors and social media variables that may affect women’s attitudes toward COVID-19 vaccination uptake have not yet been investigated. 

Regarding meta-analysis, we applied a random effects model, and performed subgroup and meta-regression analysis to overcome the high level of statistical heterogeneity. However, the limited number of studies, high heterogeneity in the results in some variables, and scarce data forced us to perform subgroup and meta-regression analysis for only a few variables. At least, the leave-one-out sensitivity analysis confirmed the robustness of our results.

## 5. Conclusions

We found that the global COVID-19 vaccination prevalence in pregnant women is low. Given the ongoing high case rates and the known increased risks of COVID-19 in pregnant women, a high vaccination rate in this vulnerable population is paramount to reducing adverse outcomes, morbidity, and mortality. An understanding of the factors related to increased COVID-19 vaccine uptake in pregnant women is essential to improve trust and increase vaccine literacy. Moreover, different public health messages and targeted information campaigns that improve COVID-19 vaccination acceptance are needed, especially in minority groups. Policy makers and healthcare professionals should reduce the fear and anxiety of pregnant women regarding the safety and efficacy of COVID-19 vaccines. Education about COVID-19 vaccines with strong and more informative messaging is important for increasing acceptance of COVID-19 vaccines among pregnant women.

## Figures and Tables

**Figure 1 vaccines-10-00766-f001:**
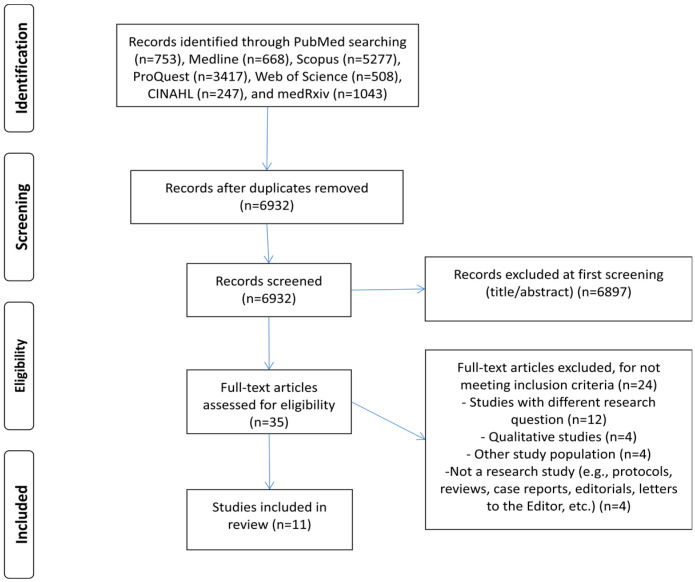
PRISMA flowchart diagram of the systematic review.

**Figure 2 vaccines-10-00766-f002:**
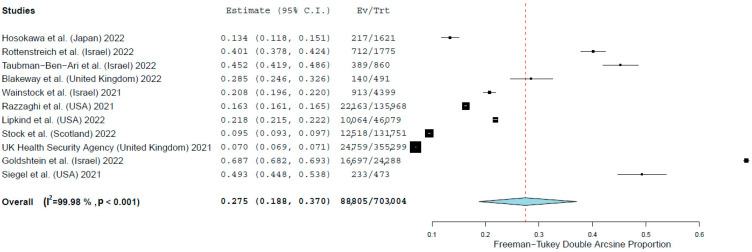
Forest plot of pregnant women vaccinated against COVID-19.

**Table 1 vaccines-10-00766-t001:** Overview of the studies included in this systematic review.

Reference	Country	Data Collection Time	Sample Size (*N*)	Age, Mean (Standard Deviation)	Study Design	Sampling Method	Response Rate (%)	COVID-19 Vaccination Uptake, % (*n*/*N*)	Publication
Hosokawa et al. [34]	Japan	28 July to 30 August 2021	1621	<29 years, 35.6%; ≥29 years, 64.4%	Cross-sectional	Convenience sampling	73.9	13.4 (217/1621)	Journal
Rottenstreich et al. [27]	Israel	19 January to 27 April 2021	1775	30.6 (5.8) for vaccinated and 29.5 (6) for unvaccinated	Cohort	NR	NR	40.2 (712/1775)	Journal
Taubman et al. [28]	Israel	March to April, 2021	860	28.3 (4.4)	Cross-sectional	Convenience sampling	65	45.2 (389/860)	Journal
Blakeway et al. [33]	United Kingdom	March to July, 2021	491	35 (NR) for vaccinated and 33 (NR) for unvaccinated	Cohort	NR	NR	28.5 (140/491)	Journal
Wainstock et al. [29]	Israel	January to June, 2021	4399	30.6 (5.3) for vaccinated and 28.2 (5.7) for unvaccinated	Cohort	NR	NR	20.8 (913/4399)	Journal
Razzaghi et al. [31]	USA	14 December 2020 to 8 May 2021	135,968	18–24 years, 13.9%; 25–34 years, 61.3%; 35–49 years, 24.8%	Cohort	NR	NR	16.3 (22,163/135,968)	Journal
Lipkind et al. [30]	USA	15 December 2020 to 22 July 2021	46,079	32.3 (4.5) for vaccinated and 29.8 (5.3) for unvaccinated	Cohort	NR	NR	21.8 (10,064/46,079)	Journal
Stock et al. [35]	Scotland	1 December 2020 to 31 October 2021	131,751	NR	Cohort	National data	NA	9.5 (12,518/131,751)	Journal
UK Health Security Agency [7]	United Kingdom	January to August 2021	355,299	NR	Cohort	National data	NA	7 (24,759/355,299)	Journal
Goldshtein et al. [26]	Israel	March to September 2021	24,288	31.6 (5.2) for vaccinated and 30.5 (5.7) for unvaccinated	Cohort	NR	NR	68.7 (16,697/24,288)	Journal
Siegel et al. [32]	USA	June to August 2021	473	33 (4.5) for vaccinated and 31.4 (5.6) for unvaccinated	Cross-sectional	Convenience sampling	69.7	49.3 (233/473)	Pre-print service

NA, not applicable; NR, not reported.

**Table 2 vaccines-10-00766-t002:** Predictors of COVID-19 vaccination uptake among pregnant women and reasons for declining vaccination.

Reference	Predictors of COVID-19 Vaccination Uptake	Reasons for Declining COVID-19 Vaccination
Blakeway et al. [33]	- Pregestational diabetes mellitus (OR = 10.5; 95% CI = 1.74 to 83.2; *p*-value = 0.014)	
Hosokawa et al. [34]		- Mistrust in government (OR = 1.26; 95% CI = 1.03 to 1.54; *p*-value = 0.001)
Razzaghi et al. [31]	- Increased age (35–49 years, 22.7%; 25–34 years, 71.8%; 18–24 years, 5.5%) - Vaccination rate was highest among Asian women (24.7%) and White women (19.7%) and lowest among Black women (6%) and Hispanic women (11.9%)	
UK Health Security [7]	- Increased age (≤24 years, 7.5% of women were vaccinated; 25–34 years, 27%; 35–44 years, 44.7%; ≥45 years, 22.1%) - Vaccination rate was highest among White women (17.5%) and Asian women (13.5%) and lowest among Black women (5.5%) - Vaccination rate was highest among women living in least deprived areas (26.5%) and lowest among women living in most deprived areas (7.8%)	
Siegel et al. [32]	- Trust in COVID-19 vaccines (OR = 6.5; 95% CI = 4.3 to 9.9; *p*-value < 0.05) - Trust in COVID-19 vaccines effectiveness for women (OR = 10.8; 95% CI = 6.7 to 17.2; *p*-value < 0.05) - Trust in COVID-19 vaccines effectiveness for newborns (OR = 6.4; 95% CI = 4.2 to 9.7; *p*-value < 0.05) - Fear of COVID-19 during pregnancy (OR = 2.5; 95% CI = 1.7 to 3.6; *p*-value < 0.05)	- Worry about safety of COVID-19 vaccines (OR = 0.16; 95% CI = 0.10 to 0.27; *p*-value < 0.05) - Worry about side effects of COVID-19 vaccines for women (OR = 0.18; 95% CI = 0.12 to 0.27; *p*-value < 0.05) - Worry about side effects of COVID-19 vaccines for newborns (OR = 0.17; 95% CI = 0.11 to 0.25; *p*-value < 0.05) - Diagnosis of COVID-19 during pregnancy (OR = 0.27; 95% CI = 0.11 to 0.69; *p*-value < 0.05)

## Data Availability

Data generated in this study are available upon reasonable request from the first author, P.G.

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
