# Peer review of "Uptake of COVID-19 Vaccines among Pregnant Women: A Systematic Review and Meta-Analysis"

_vaccines, 2022, doi:10.3390/vaccines10050766_

Round 1

Reviewer 1 Report

The work is of interest to both administrative authorities and professional researchers, and therefore, it deserves  publication.

A minor issue is that the authors should clearly clarify that the COVID-19 vaccines are commonly officially suggested for vaccination in pregnant women.  

Author Response

Dear Reviewer, We would like to thank you for your time and your invaluable comments regarding our manuscript. We read the suggestions and concerns from you and we have addressed all of them carefully. We modified and revised our manuscript as suggested.

Your comment:
A minor issue is that the authors should clearly clarify that the COVID-19 vaccines are commonly officially suggested for vaccination in pregnant women.

Our answer:
Done. We added a sentence in page 2 to make clear that the COVID-19 vaccines are commonly officially suggested for vaccination in pregnant women. 

Reviewer 2 Report

The systematic review and meta-analysis (manuscript ID: vaccines-1697116) entitled “Uptake of COVID-19 vaccines among pregnant women: a systematic review and meta-analysis” by Dr. Galanis evaluated comprehensively the most recent literature for estimating the overall proportion of vaccinated pregnant women against the COVID-19. The authors performed a meta-analysis of 11 studies including 703,004 pregnant women. Main results indicate that the overall proportion of vaccinated pregnant women against the COVID-19 was 27.5%, while predictors of COVID-19 vaccination uptake were older age, ethnicity, race, trust in COVID-19 vaccines, and fear of COVID-19 during pregnancy. The present manuscript is well written. The analysis is well performed. Figures are highly informative. I therefore recommend a minor revision.  The analysis will improve our knowledge on COVID-19 vaccination programs. The ms can be accepted following a minor revision. I have few suggestions

GENERAL COMMENTS
Despite the authors stated that this is the first systematic review which provide evidence about the uptake of COVID-19 vaccines among pregnant women, an additional, recent similar study has been published on Vaccines (https://fanyv88.com:443/https/doi.org/10.3390/vaccines10020246). The analysis was conducted on 6 studies. This work should therefore be included and disucssed. The author should compare their results with those already reported in the meta-analysis, in the discussion. Several sentences should also be modified accordingly, such as lines 199-200, as well as others

Table 1 should be reorganized for a better readability. 

Why don’t use also SARS-CoV-2 other than covid-19 as query in the research strategy in PubMed?

It might be interesting to evaluate the vaccine dose of women included in the analysis, .e., first, second third or booster dose. Have the authors evaluated this such of variable? 

MINOR
Lines 33-39 As correctly stated, COVID-19 were more likely to have iatrogenic preterm births. In general, during pregnancy, pregnant females are prone to develop an higher susceptibility to viral infections, which might lead, in certain circumstances, to pregnancy complications, preterm births and even pregnancy losses https://fanyv88.com:443/https/doi.org/10.3390/vaccines8030473 and https://fanyv88.com:443/https/doi.org/10.1093/humupd/dmv041. This information and references should be included as a support. 

Author Response

Dear Reviewer, We would like to thank you for your time and your invaluable comments regarding our manuscript. We read the suggestions and concerns from you and we have addressed all of them carefully. We modified and revised our manuscript as suggested. We made a revision which we hope meet with your approval.Please find below, a point to point response to your comments and concerns. In our manuscript, we marked the revisions with track changes system.

GENERAL COMMENTS

Despite the authors stated that this is the first systematic review which provide evidence about the uptake of COVID-19 vaccines among pregnant women, an additional, recent similar study has been published on Vaccines (https://fanyv88.com:443/https/doi.org/10.3390/vaccines10020246). The analysis was conducted on 6 studies. This work should therefore be included and disucssed. The author should compare their results with those already reported in the meta-analysis, in the discussion. Several sentences should also be modified accordingly, such as lines 199-200, as well as others

Answer: Done. Dear Reviewer, we added this reference and we discussed it. Please, see the last paragraph before limitations in the Discussion section.

Table 1 should be reorganized for a better readability. 

Answer: Done. Please, see Table 1.

Why don’t use also SARS-CoV-2 other than covid-19 as query in the research strategy in PubMed?

Answer: We considered that the search term “covid-19” would be fair enough to find out all relevant studies, but you are right. It would be better to include also the term “SARS-CoV-2”.

It might be interesting to evaluate the vaccine dose of women included in the analysis, .e., first, second third or booster dose. Have the authors evaluated this such of variable? 

Answer: Dear Reviewer, you are right but unfortunately most of the studies included in our review did not include this information. Thus we decided not to include this information.

MINOR
Lines 33-39 As correctly stated, COVID-19 were more likely to have iatrogenic preterm births. In general, during pregnancy, pregnant females are prone to develop an higher susceptibility to viral infections, which might lead, in certain circumstances, to pregnancy complications, preterm births and even pregnancy losses https://fanyv88.com:443/https/doi.org/10.3390/vaccines8030473 and https://fanyv88.com:443/https/doi.org/10.1093/humupd/dmv041. This information and references should be included as a support. 

Answer: Done. Dear Reviewer, we added your comment and the two references. Please, see the first paragraph in the Introduction section.